# Discrete Versions of Jensen–Fisher, Fisher and Bayes–Fisher Information Measures of Finite Mixture Distributions

**DOI:** 10.3390/e23030363

**Published:** 2021-03-18

**Authors:** Omid Kharazmi, Narayanaswamy Balakrishnan

**Affiliations:** 1Department of Statistics, Faculty of Mathematical Sciences, Vali-e-Asr University of Rafsanjan, P.O. Box 518, Rafsanjan, Iran; Omid.kharazmi@vru.ac.ir; 2Department of Mathematics and Statistics, McMaster University, Hamilton, ON L8S 4K1, Canada

**Keywords:** Fisher information, Bayes–Fisher information, chi-square divergence, Kullback–Leibler divergence, Jensen–Shannon entropy

## Abstract

In this work, we first consider the discrete version of Fisher information measure and then propose Jensen–Fisher information, to develop some associated results. Next, we consider Fisher information and Bayes–Fisher information measures for mixing parameter vector of a finite mixture probability mass function and establish some results. We provide some connections between these measures with some known informational measures such as chi-square divergence, Shannon entropy, Kullback–Leibler, Jeffreys and Jensen–Shannon divergences.

## 1. Introduction

Over the last seven decades, several different criteria have been introduced in the literature for measuring uncertainty in a probabilistic model. Shannon entropy and Fisher information are the most important information measures that have been used rather extensively. Information theory started with Shannon entropy, introduced in the pioneering work of Shannon [1], based on a study of systems described by probability density (or mass) functions. About two decades earlier, Fisher [2] had proposed another information measure, describing the interior properties of a probabilistic model, that plays an important role in likelihood-based inferential methods. Fisher information and Shannon entropy are fundamental criteria in statistical inference, physics, thermodynamics and information theory. Complex systems can be described by means of their behavior (Shannon) and their architecture (Fisher) information. For more discussions, see Zegers [3] and Balakrishnan and Stepanov [4].

Let *X* be a discrete random variable with probability mass function (PMF) $P=(p_{1},\ldots ,p_{n})$. Then, the Shannon entropy of random variable *X* is defined as
$$H\left(X\right)=H\left(P\right)=-\sum_{i=1}^{n}p_{i}\log p_{i},$$
where “log” denotes the natural logarithm. For more details, see Shannon [1]. Following the work of Shannon [1], considerable attention has been paid to providing some extensions of Shannon entropy. Jensen–Shannon (JS) divergence is one such important extension of Shannon entropy that has been widely used; see Lin [5]. The Jensen–Shannon divergence between two probability mass functions $P=(p_{1},p_{2},\ldots ,p_{n})$ and $Q=(q_{1},q_{2},\ldots ,q_{n})$, for 0≤α≤1, is defined as
$$\begin{array}{c} JS(P,Q;\alpha )=H(\alpha P+(1-\alpha )Q)-\alpha H(P)-(1-\alpha )H(Q). \end{array}$$

The JS divergence is a smoothed and symmetric version of the most important divergence measure of information theory, namely, Kullback–Leibler divergence. Recently, Jensen–Fisher (JF) and Jensen–Gini (JG) divergence measures have been introduced by Sánchez-Moreno et al. [6] and Mehrali et al. [7], respectively.

In the present paper, motivated by the idea of JS divergence, we consider discrete versions of Fisher information (DFI) and Fisher information distance (DFID), and then develop a new information measure associated with DFI measure. In addition, we provide some results for the Fisher information of a finite mixture probability mass function through a Bayesian perspective. The discrete Fisher information of a random variable *X* with PMF $P=(p_{1},p_{2},\ldots ,p_{n})$ is defined as
(1)$$\begin{array}{c} I\left(P\right)=\sum_{i=1}^{n}\frac{{\left(p_{i+1}-p_{i}\right)}^{2}}{p_{i}}, \end{array}$$
with $p_{n+1}=0$.

The Fisher information in (Equation 1) has been made use of in the processing of complex and stationary signals. For example, the discrete version of Fisher information has been used in detecting epileptic seizures in EEG signals recorded in humans and turtles, in detecting dynamical changes in many non-linear models such as logistic map and Lorenz model, and also in the analysis of geoelectrical signals; see Martin et al. [8], Ramírez-Pacheco et al. [9] and Ramírez-Pacheco et al. [10] for pertinent details.

The discrete Fisher information distance (DFID) between two probability mass functions $P=(p_{1},p_{2},\ldots ,p_{n})$ and $Q=(q_{1},q_{2},\ldots ,q_{n})$ is defined as
(2)$$\begin{array}{c} D\left(P,Q\right)=\sum_{i=1}^{n}{\left(\frac{p_{i+1}}{p_{i}}-\frac{q_{i+1}}{q_{i}}\right)}^{2}p_{i}, \end{array}$$
where, as above, $p_{n+1}=q_{n+1}=0$. For some of its properties, one may refer to Ramírez-Pacheco et al. [10] and Johnson [11].

With regard to informational properties of finite mixture models, one may refer to Contreras-Reyes and Cortés [12] and Abid et al. [13]. These authors have provided upper and lower bounds for Shannon and Rényi entropies of non-gaussian finite mixtures, skew-normal and skew-t distributions, respectively. Kolchinsky and Tracey [14] have studied the upper and lower bounds for the entropy of Gaussian mixture distributions using the Bhattacharyya and Kullback–Leibler divergences.

The first purpose of this paper is to propose Jensen–Fisher information for discrete random variables $X_{1},\ldots ,X_{n}$, with probability mass functions $P_{1},\ldots ,P_{n},$ respectively. For this purpose, we first define discrete version of Jensen–Fisher information for two PMFs P and Q, and then provide some results concerning this new information measure. Then, this idea is extended to the general case of PMFs $P_{1},\ldots ,P_{n}$.

The second purpose of this work is to study Fisher and Bayes–Fisher information measures for the mixing parameter of a finite mixture probability mass function. Let $P_{1},\ldots ,P_{n}$ be *n* probability mass functions, where $P_{j}=\left(p_{j1},\ldots ,p_{jk}\right)$. Then, a finite mixture probability mass function with mixing parameter vector $\theta =(\theta_{1},\ldots ,\theta_{n-1})$, for n≥2, is given by $P_{\theta }=\left(p_{\theta }^{1},\ldots ,p_{\theta }^{k}\right),$ where
(3)$$\begin{array}{c} p_{\theta }^{j}=\frac{1}{n-1}\sum_{i=1}^{n-1}\theta_{i}p_{ij}+\left(1-\frac{\sum_{i=1}^{n-1}\theta_{i}}{n-1}\right)p_{nj},\;j=1,\ldots ,k, \end{array}$$
$0\leq \theta_{i}\leq 1,\;i=1,\ldots ,n-1$ and $\sum_{i=1}^{n-1}\theta_{i}\leq 1$.

Let *X* and *Y* be two discrete random variables with PMFs $P=(p_{1},\ldots ,p_{n})$ and $Q=(q_{1},\ldots ,q_{n})$, respectively. Then, the Kullback–Leibler (KL) distance between *X* and *Y* (or P and Q) is defined as
$$KL\left(X||Y\right)=KL\left(P,Q\right)=\sum_{i=1}^{n}p_{i}\log\left(\frac{p_{i}}{q_{i}}\right).$$
The Kullback–Leibler discrimination between *Y* and *X* can be defined similarly. For more details, see Kullback and Leibler [15]. The chi-square divergence between PMFs P and Q is defined by
$$\begin{array}{c} \;\chi^{2}\left(P,Q\right)=\sum_{i=1}^{n}\frac{{\left(p_{i}-q_{i}\right)}^{2}}{p_{i}}. \end{array}$$
For pertinent details, see Broniatowski [16] and Cover and Thomas [17].

The rest of this paper is organized as follows. In Section 2, we first consider discrete version of Fisher information and then propose the discrete Jensen–Fisher information (DJFI) measure. We show that DJFI measure can be represented based on the mixture of discrete Fisher information distance measures. In Section 3, we consider a finite mixture probability mass function and establish some results for the Fisher information measure of the mixing parameter vector. We show that the Fisher information of the mixing parameter vector is connected to chi-square divergence. Next, in Section 4, we discuss the Bayes–Fisher information for the mixing parameter vector of probability mass functions under some prior distributions for the mixing parameter. We then show that this measure is connected to Shannon entropy, Jensen–Shannon entropy, Kullback–Leibler and Jeffreys divergence measures. Finally, we present some concluding remarks in Section 5.

## 2. Discrete Version of Jensen-Fisher Information

In this section, we first give a result for the DFI measure based on the log-convex and log-concave property of the probability mass function. Then, we define the discrete Jensen–Fisher information measure, and establish some interesting properties of it.

**Theorem** **1.**
*Let $P=(p_{1},p_{2},\ldots ,p_{n})$ be a probability mass function.*
*(i)* 
*If P is log-concave, then $I\left(P\right)\leq p_{1};$*
*(ii)* 
*If P is log-convex, then $I\left(P\right)\geq p_{1}.$*



**Proof.** P is log-convex (log-concave) if $p_{i}^{2}\geq \left(\leq \right)p_{i-1}p_{i+1}\;\forall i.$ So, from the definition of DFI in (Equation 1), we have
$$\begin{array}{ccc} I(P) & = & \sum_{i=1}^{n}\frac{{\left(p_{i+1}-p_{i}\right)}^{2}}{p_{i}}\geq \left(\leq \right)p_{1}. \end{array}$$
  □

### 2.1. Discrete Jensen–Fisher Information Based on Two Probability Mass Functions P and Q

We first define a symmetric version of DFID measure in (Equation 2), and then propose the discrete Jensen–Fisher information measure involving two probability mass functions.

**Definition** **1.**
*Let P and Q be two probability mass functions given by $P=(p_{1},p_{2},\ldots ,p_{n})$ and $Q=(q_{1},q_{2},\ldots ,q_{n})$. Then, a symmetric version of discrete Fisher information distance in (Equation 2) is defined as*
$$\begin{array}{ccc} \mathrm{SD}(P,Q) & = & \frac{1}{2}D\left(P,\frac{P+Q}{2}\right)+\frac{1}{2}D\left(Q,\frac{P+Q}{2}\right). \end{array}$$


**Definition** **2.**
*Let P and Q be two probability mass functions given by $P=(p_{1},p_{2},\ldots ,p_{n})$ and $Q=(q_{1},q_{2},\ldots ,q_{n}).$ Then, the discrete Jensen–Fisher information is defined as*
(4)$$\begin{array}{ccc} \mathrm{JFI}(P,Q) & = & \frac{I(P)+I(Q)}{2}-I\left(\frac{P+Q}{2}\right). \end{array}$$


In the following theorem, we show that the discrete Jensen–Fisher information measure can be obtained based on mixtures of Fisher information distances.

**Theorem** **2.**
*Let P and Q be two probability mass functions given by $P=(p_{1},p_{2},\ldots ,p_{n})$ and $Q=(q_{1},q_{2},\ldots ,q_{n})$. Then,*
$$\begin{array}{ccc} \mathrm{JFI}(P,Q) & = & \frac{1}{2}D\left(P,\frac{P+Q}{2}\right)+\frac{1}{2}D\left(Q,\frac{P+Q}{2}\right)\\ & = & \mathrm{SD}(P,Q). \end{array}$$


**Proof.** From the definition of DFID in (Equation 2), we get
$$\begin{array}{ccc} D\left(P,\frac{P+Q}{2}\right) & = & \sum_{i=1}^{n}{\left(\frac{p_{i+1}}{p_{i}}-\frac{p_{i+1}+q_{i+1}}{p_{i}+q_{i}}\right)}^{2}p_{i}\\ & = & \sum_{i=1}^{n}{\left\{\left(\frac{p_{i+1}}{p_{i}}-1\right)-\left(\frac{p_{i+1}+q_{i+1}}{p_{i}+q_{i}}-1\right)\right\}}^{2}p_{i}\\ & = & \sum_{i=1}^{n}{\left(\frac{p_{i+1}}{p_{i}}-1\right)}^{2}p_{i}-2\sum_{i=1}^{n}\left(\frac{p_{i+1}}{p_{i}}-1\right)\left(\frac{p_{i+1}+q_{i+1}}{p_{i}+q_{i}}-1\right)p_{i}\\ & & +\sum_{i=1}^{n}{\left(\frac{p_{i+1}+q_{i+1}}{p_{i}+q_{i}}-1\right)}^{2}p_{i}\\ & = & \sum_{i=1}^{n}\frac{{\left(p_{i+1}-p_{i}\right)}^{2}}{p_{i}}-2\sum_{i=1}^{n}\left(p_{i+1}-p_{i}\right)\frac{\left(p_{i+1}+q_{i+1}-\left(p_{i}+q_{i}\right)\right)}{p_{i}+q_{i}}\\ & & +\sum_{i=1}^{n}\frac{{\left(p_{i+1}+q_{i+1}-\left(p_{i}+q_{i}\right)\right)}^{2}}{{\left(p_{i}+q_{i}\right)}^{2}}p_{i}. \end{array}$$
In a similar way, we get
$$\begin{array}{ccc} D\left(Q,\frac{P+Q}{2}\right) & = & \sum_{i=1}^{n}\frac{{\left(q_{i+1}-q_{i}\right)}^{2}}{q_{i}}-2\sum_{i=1}^{n}\left(q_{i+1}-q_{i}\right)\frac{\left(p_{i+1}+q_{i+1}-\left(p_{i}+q_{i}\right)\right)}{p_{i}+q_{i}}\\ & & +\sum_{i=1}^{n}\frac{{\left(p_{i+1}+q_{i+1}-\left(p_{i}+q_{i}\right)\right)}^{2}}{{\left(p_{i}+q_{i}\right)}^{2}}q_{i}. \end{array}$$
Upon adding the above two expressions, we obtain
$$\begin{array}{ccc} D\left(P,\frac{P+Q}{2}\right)+D\left(Q,\frac{P+Q}{2}\right) & = & \sum_{i=1}^{n}\frac{{\left(p_{i+1}-p_{i}\right)}^{2}}{p_{i}}+\sum_{i=1}^{n}\frac{{\left(q_{i+1}-q_{i}\right)}^{2}}{q_{i}}\\ & & -\sum_{i=1}^{n}\frac{{\left(p_{i+1}+q_{i+1}-\left(p_{i}+q_{i}\right)\right)}^{2}}{p_{i}+q_{i}}\\ & = & I\left(P\right)+I\left(Q\right)-2I\left(\frac{P+Q}{2}\right)\\ & = & 2\mathrm{JFI}(P,Q), \end{array}$$
as required.  □

**Example** **1.**
*Let*
$$X=\left\{\begin{array}{cc} 1, & with\;probability\;p,\\ 0, & with\;probability\;1-p, \end{array}\right.$$
*and*
$$Y=\left\{\begin{array}{cc} 1, & with\;probability\;q,\\ 0, & with\;probability\;1-q. \end{array}\right.$$


The corresponding PMFs of variables *X* and *Y* are given by P=(p,1−p) and Q=(q,1−q), respectively. From Theorem 2, we then have
$$\begin{array}{ccc} \mathrm{JFI}(P,Q) & = & \frac{p+q}{2}{\left(\frac{1-p}{p}-\frac{1-q}{q}\right)}^{2}. \end{array}$$
A 3D-plot of this JFI(P,Q) is presented in Figure 1.

### 2.2. Discrete Jensen–Fisher Information Based on *n* Probability Mass Functions $P_{1},\ldots ,P_{n}$


Let $P_{1},\ldots ,P_{n}$ be *n* probability mass functions, where $P_{i}=\left(p_{i1},\ldots ,p_{ik}\right)$. In the following definition, we extend the discrete Jensen–Fisher information measure in (Equation 4) to the case of *n* probability mass functions.

**Definition** **3.**
*Let $P_{1},\ldots ,P_{n}$ be n probability mass functions given by $P_{i}=\left(p_{i1},p_{i2},\ldots ,p_{ik}\right)$, i=1,2,…,n, with $\sum_{j=1}^{k}p_{ij}=1$, and $\alpha_{1},\ldots ,\alpha_{n}$ be non-negative real numbers such that $\sum_{i=1}^{n}\alpha_{i}=1$. Then, the discrete Jensen–Fisher information (DJFI) based on the n probability mass functions is defined as*
(5)$$\begin{array}{ccc} \mathrm{JFI}(P_{1},\ldots ,P_{n};\underline{\alpha }) & = & \sum_{i=1}^{n}\alpha_{i}I\left(P_{i}\right)-I\left(\sum_{i=1}^{n}\alpha_{i}P_{i}\right)\\ \; & = & \sum_{i=1}^{n}\alpha_{i}\sum_{j=1}^{k}\frac{{\left(p_{ij+1}-p_{ij}\right)}^{2}}{p_{ij}}-\sum_{j=1}^{k}\frac{{\left(\sum_{i=1}^{n}\alpha_{i}p_{ij+1}-\sum_{i=1}^{n}\alpha_{i}p_{ij}\right)}^{2}}{\sum_{i=1}^{n}\alpha_{i}p_{ij}}, \end{array}$$
*where $\underline{\alpha }=\left(\alpha_{1},\ldots ,\alpha_{n}\right).$*


**Theorem** **3.**
*Let $P_{1},\ldots ,P_{n}$ be n probability mass functions given by $P_{i}=\left(p_{i1},p_{i2},\ldots ,p_{ik}\right)$, i=1,2,…,n, and $\sum_{j=1}^{k}p_{ij}=1$. Then, the DJFI measure can be expressed as a mixture of DFID measures in (Equation 2) as follows:*
$$\begin{array}{ccc} \mathrm{JFI}(P_{1},\ldots ,P_{n},\underline{\alpha }) & = & \sum_{i=1}^{n}\alpha_{i}D\left(P_{i},P_{T}\right), \end{array}$$
*where $P_{T}=\sum_{i=1}^{n}\alpha_{i}P_{i}$ is the weighted PMF.*


**Proof.** From the definition in (Equation 5), we get
$$\begin{array}{ccc} \sum_{i=1}^{n}\alpha_{i}D\left(P_{i},P_{T}\right) & = & \sum_{i=1}^{n}\alpha_{i}\sum_{j=1}^{k}{\left(\frac{p_{ij+1}}{p_{ij}}-\frac{\sum_{i=1}^{n}\alpha_{i}p_{ij+1}}{\sum_{i=1}^{n}\alpha_{i}p_{ij}}\right)}^{2}p_{ij}\\ & = & \sum_{i=1}^{n}\alpha_{i}\sum_{j=1}^{k}{\left\{\frac{p_{ij+1}}{p_{ij}}-1-\left(\frac{\sum_{i=1}^{n}\alpha_{i}p_{ij+1}}{\sum_{i=1}^{n}\alpha_{i}p_{ij}}-1\right)\right\}}^{2}p_{ij}\\ & = & \sum_{i=1}^{n}\alpha_{i}\sum_{j=1}^{k}\frac{{\left(p_{ij+1}-p_{ij}\right)}^{2}}{p_{ij}}-2\sum_{j=1}^{k}\frac{{\left(\sum_{i=1}^{n}\alpha_{i}p_{ij+1}-\sum_{i=1}^{n}\alpha_{i}p_{ij}\right)}^{2}}{\sum_{i=1}^{n}\alpha_{i}p_{ij}}\\ & & +\sum_{j=1}^{k}\frac{{\left(\sum_{i=1}^{n}\alpha_{i}p_{ij+1}-\sum_{i=1}^{n}\alpha_{i}p_{ij}\right)}^{2}}{\sum_{i=1}^{n}\alpha_{i}p_{ij}}\\ & = & \sum_{i=1}^{n}\alpha_{i}\sum_{j=1}^{k}\frac{{\left(p_{ij+1}-p_{ij}\right)}^{2}}{p_{ij}}-\sum_{j=1}^{k}\frac{{\left(\sum_{i=1}^{n}\alpha_{i}p_{ij+1}-\sum_{i=1}^{n}\alpha_{i}p_{ij}\right)}^{2}}{\sum_{i=1}^{n}\alpha_{i}p_{ij}}\\ & = & \sum_{i=1}^{n}\alpha_{i}I\left(P_{i}\right)-I\left(\sum_{i=1}^{n}\alpha_{i}P_{i}\right)\\ & = & \mathrm{JFI}(P_{1},\ldots ,P_{n},\underline{\alpha }), \end{array}$$
as required. □

## 3. Fisher Information of a Finite Mixture Probability Mass Function

In this section, we discuss Fisher information for parameter θ of a finite mixture probability mass function.

**Theorem** **4.**
*The Fisher information of PMF in (Equation 3) about parameter $\theta_{i},i=1,\ldots ,n-1$, is given by*
(6)$$\begin{array}{c} I\left(\theta_{i}\right)=\frac{1}{{\left(\theta_{i}-\left(n-1\right)\right)}^{2}}\chi^{2}\left(P_{\theta_{-i}},P_{\theta }\right),\;i=1,\ldots ,n-1, \end{array}$$
*where $P_{\theta_{-i}}=\left(p_{\theta_{-i}}^{1},\ldots ,p_{\theta_{-i}}^{k}\right),$*
$$p_{\theta_{-i}}^{j}=\frac{n-2}{n-1}p_{ij}+\frac{1}{n-1}\sum_{ł=1,ł\neq i}^{n-1}\theta_{ł}p_{łj}+\frac{1}{n-1}\left(1-\sum_{ł=1,ł\neq i}^{n-1}\theta_{ł}\right)p_{nj},\;j=1,\ldots ,k,$$
*and $\theta_{-i}=\left(\theta_{1},\ldots ,\theta_{i-1},\theta_{i+1},\ldots ,\theta_{n-1}\right).$*


**Proof.** From the definition of Fisher information in (Equation 1) and for i=1,…,n−1, we have
(7)$$\begin{array}{ccc} I(\theta_{i}) & = & \sum_{j=1}^{k}{\left[\frac{\partial \log P_{\theta }}{\partial \theta_{i}}\right]}^{2}p_{\theta }^{j}\\ \; & = & \frac{1}{{\left(n-1\right)}^{2}}\sum_{j=1}^{k}\frac{{\left(p_{ij}-p_{nj}\right)}^{2}}{{\left(p_{\theta }^{j}\right)}^{2}}p_{\theta }^{j}\\ \; & = & \frac{1}{{\left(\theta_{i}-\left(n-2\right)\right)}^{2}}\sum_{j=1}^{k}\frac{(p_{\theta_{-i}}^{j}-p_{\theta }^{j}{)}^{2}}{p_{\theta }^{j}}\\ \; & = & \frac{1}{{\left(\theta_{i}-\left(n-1\right)\right)}^{2}}\chi^{2}\left(P_{\theta_{-i}},P_{\theta }\right),\;i=1,\ldots ,n-1, \end{array}$$
where the third equation follows from the fact that, for i=1,…,n−1,
$$p_{ij}-p_{nj}=\frac{n-1}{\theta_{i}-\left(n-2\right)}\left(p_{\theta }^{j}-p_{\theta_{-i}}^{j}\right).$$
  □

## 4. Bayes–Fisher Information of a Finite Mixture Probability Mass Function

In this section, we discuss Bayes–Fisher information for the mixing parameter vector θ of the finite mixture probability mass function in (Equation 3) under some prior distributions for the mixing parameter vector. We now introduce two notations that will be used in the sequel. Consider the parameter vector $\theta =(\theta_{1},\ldots ,\theta_{n-1})$, and then define $\left(0_{i},\theta \right)=\left(\theta_{1},\ldots ,\theta_{i-1},0,\theta_{i+1},\ldots ,\theta_{n-1}\right)$ and $\left(1_{i},\theta \right)=\left(\theta_{1},\ldots ,\theta_{i-1},1,\theta_{i+1},\ldots ,\theta_{n-1}\right).$

**Theorem** **5.**
*The Bayes–Fisher information for parameter $\theta_{i},\;i=1,\ldots ,n-1$, of the finite mixture PMF in (Equation 3), under the uniform prior on [0,1], is given by*
$$\begin{array}{ccc} \tilde{I}\left(\theta_{i}\right) & = & KL\left(P_{\left(1_{i},\theta \right)},P_{\left(0_{i},\theta \right)}\right)+KL\left(P_{\left(0_{i},\theta \right)},P_{\left(1_{i},\theta \right)}\right)\\ & = & J\left(P_{\left(0_{i},\theta \right)},P_{\left(1_{i},\theta \right)}\right), \end{array}$$
*where $P_{\left(1_{i},\theta \right)}=\left(p_{\left(1_{i},\theta \right)}^{1},\ldots ,p_{\left(1_{i},\theta \right)}^{n}\right),$ with*
(8)$$p_{\left(1_{i},\theta \right)}^{j}=\frac{1}{n-1}p_{ij}+\frac{1}{n-1}\sum_{ł=1,ł\neq i}^{n-1}\theta_{ł}+\left(1-\frac{1}{n-1}\left(1+\sum_{ł=1,ł\neq i}^{n-1}\theta_{ł}\right)\right)p_{nj},$$
*and $P_{\left(0_{i},\theta \right)}=\left(p_{\left(0_{i},\theta \right)}^{1},\ldots ,p_{\left(0_{i},\theta \right)}^{n}\right),$ with*
(9)$$p_{\left(0_{i},\theta \right)}^{j}=\frac{1}{n-1}\sum_{ł=1,ł\neq i}^{n-1}\theta_{ł}p_{łj}+\left(1-\frac{1}{n-1}\sum_{ł=1,ł\neq i}^{n-1}\theta_{ł}\right)p_{nj},$$
*and J corresponds to Jeffreys’ divergence.*


**Proof.** By definition and from (Equation 7), for i=1,…,n−1, we have
(10)$$\begin{array}{ccc} \tilde{I}\left(\theta_{i}\right) & = & E\left[I\left(\Theta_{i}\right)\right]=\frac{1}{{\left(n-1\right)}^{2}}\int_{0}^{1}\left\{\sum_{j=1}^{k}\frac{{\left(p_{ij}-p_{nj}\right)}^{2}}{p_{\theta }^{j}}\right\}d\theta_{i}\\ \; & = & \frac{1}{n-1}\sum_{j=1}^{k}\left(p_{ij}-p_{nj}\right)\left\{\int_{0}^{1}\frac{1}{n-1}\frac{p_{ij}-p_{nj}}{p_{\theta }^{j}}d\theta_{i}\right\}\\ \; & = & \frac{1}{n-1}\sum_{j=1}^{k}\left(p_{ij}-p_{nj}\right)\left\{\log\left(p_{\theta }^{j}\right){|}_{0}^{1}\right\}. \end{array}$$
On the other hand, we have
(11)$$\begin{array}{c} p_{\left(1_{i},\theta \right)}-p_{\left(0_{i},\theta \right)}=\frac{1}{n-1}\left(p_{ij}-p_{nj}\right). \end{array}$$
Hence, upon substituting (Equation 11) into (Equation 10), we obtain
$$\begin{array}{ccc} \tilde{I}\left(\theta_{i}\right) & = & \frac{1}{n-1}\sum_{j=1}^{k}\left(p_{ij}-p_{nj}\right)\log\left\{\frac{p_{\left(1_{i},\theta \right)}^{j}}{p_{\left(0_{i},\theta \right)}^{j}}\right\}\\ & = & \sum_{j=1}^{k}\left(p_{\left(1_{i},\theta \right)}-p_{\left(0_{i},\theta \right)}\right)\log\left\{\frac{p_{\left(1_{i},\theta \right)}^{j}}{p_{\left(0_{i},\theta \right)}^{j}}\right\}\\ & = & KL\left(P_{\left(1_{i},\theta \right)},P_{\left(0_{i},\theta \right)}\right)+KL\left(P_{\left(0_{i},\theta \right)},P_{\left(1_{i},\theta \right)}\right)\\ & = & J\left(P_{\left(0_{i},\theta \right)},P_{\left(1_{i},\theta \right)}\right), \end{array}$$
as required.  □

**Theorem** **6.**
*For the mixture model with PMF in (Equation 3), we have the following:*
*(i)* 
*The Bayes–Fisher information for $\theta_{i},i=1,\ldots ,n-1$, under Beta(2,1) prior with PMF $\pi \left(\theta_{i}\right)=2\theta_{i},\theta_{i}\in \left[0,1\right]$, is*
$$\begin{array}{ccc} \tilde{I}\left(\theta_{i}\right) & = & 2KL(P_{\left(0_{i},\theta \right)},P_{\left(1_{i},\theta \right)}),\;i=1,\ldots ,n-1; \end{array}$$
*(ii)* 
*The Bayes-Fisher information for parameter $\theta_{i},i=1,\ldots ,n-1$, under Beta(1,2) prior with PMF $\pi \left(\theta_{i}\right)=2\left(1-\theta_{i}\right),\theta_{i}\in \left[0,1\right]$, is*
$$\begin{array}{ccc} \tilde{I}\left(\theta_{i}\right) & = & 2KL(P_{\left(1_{i},\theta \right)},P_{\left(0_{i},\theta \right)}),i=1,\ldots ,n-1. \end{array}$$



**Proof.** By definition, and from (Equation 7), for i=1,…,n−1, we have
$$\begin{array}{ccc} \tilde{I}\left(\theta_{i}\right) & = & E\left[I\left(\Theta_{i}\right)\right]=\frac{1}{{\left(n-1\right)}^{2}}\int_{0}^{1}\left\{\sum_{j=1}^{k}\frac{{\left(p_{ij}-p_{nj}\right)}^{2}}{p_{\theta }^{j}}\right\}\pi \left(\theta_{i}\right)d\theta_{i}\\ & = & \frac{2}{n-1}\sum_{j=1}^{k}\left(p_{ij}-p_{nj}\right)\left\{\int_{0}^{1}\frac{\theta_{i}}{n-1}\frac{p_{ij}-p_{nj}}{p_{\theta }^{j}}d\theta_{i}\right\}\\ & = & \frac{2}{n-1}\sum_{j=1}^{k}\left(p_{ij}-p_{nj}\right)\left\{\int_{0}^{1}\frac{p_{\theta }^{j}-p_{\left(0_{i},\theta \right)}^{j}}{p_{\theta }\left(x\right)}d\theta_{i}\right\}\\ & = & \frac{2}{n-1}\sum_{j=1}^{k}\left(p_{ij}-p_{nj}\right)\left\{1-\frac{\left(n-1\right)p_{\left(0_{i},\theta \right)}^{j}}{p_{ij}-p_{nj}}\log\left(p_{\theta }^{j}\right){|}_{0}^{1}\right\}\\ & = & -2\sum_{j=1}^{k}p_{\left(0_{i},\theta \right)}^{j}\log\left\{\frac{p_{\left(1_{i},\theta \right)}^{j}}{p_{\left(0_{i},\theta \right)}^{j}}\right\}\\ & = & 2\sum_{j=1}^{k}p_{\left(0_{i},\theta \right)}^{j}\log\left\{\frac{p_{\left(0_{i},\theta \right)}^{j}}{p_{\left(1_{i},\theta \right)}^{j}}\right\}\\ & = & 2KL(P_{\left(0_{i},\theta \right)},P_{\left(1_{i},\theta \right)}), \end{array}$$
as required for Part (i). Part (ii) can be proved in an analogous manner.Let us now consider the following general triangular prior for the parameter $\theta_{i},i=1,\ldots ,n-1$:
(12)$$\pi_{\alpha }\left(\theta_{i}\right)=\left\{\begin{array}{cc} \frac{2\theta_{i}}{\alpha }, & 0<\theta_{i}\leq \alpha ,\\ \frac{2(1-\theta_{i})}{1-\alpha }, & \alpha \leq \theta_{i}<1, \end{array}\right.$$
for some α∈(0,1).  □

**Theorem** **7.**
*The Bayes–Fisher information for parameter $\theta_{i},i=1,\ldots ,n-1,$ with the general triangular prior with density $\pi_{\alpha }\left(\theta_{i}\right)$ in (Equation 12), is given by*
$$\begin{array}{ccc} \tilde{I}\left(\theta_{i}\right) & = & \frac{2}{\alpha (1-\alpha )}\left[\alpha KL\left(P_{\left(1_{i},\theta \right)},P_{\alpha }\right)+\left(1-\alpha \right)KL\left(P_{\left(0_{i},\theta \right)},P_{\alpha }\right)\right]\\ & = & \frac{2}{\alpha (1-\alpha )}JS\left(P_{\left(0_{i},\theta \right)},P_{\left(1_{i},\theta \right)};\alpha \right), \end{array}$$
*where $P_{\alpha }=\left(p_{\alpha }^{1},\ldots ,p_{\alpha }^{k}\right)$ is a finite mixture PMF, with*
$$\begin{array}{ccc} p_{\alpha }^{j} & = & \frac{\alpha }{n-1}p_{ij}+\frac{1}{n-1}\sum_{l=1,l\neq i}^{n-1}\theta_{l}p_{lj}+\left(1-\frac{1}{n-1}\left(\alpha +\sum_{l=1,l\neq i}^{n-1}\theta_{l}\right)\right)p_{nj} \end{array}$$
*and $P_{\left(1_{i},\theta \right)}$ and $P_{\left(0_{i},\theta \right)}$ are as defined in (Equation 8) and (Equation 9), respectively.*


**Proof.** From the assumptions made, for i=1,…,n−1, we have
$$\begin{array}{ccc} \tilde{I}\left(\theta_{i}\right) & = & E\left[I(\Theta_{i})\right]=\int_{0}^{\alpha }I\left(\theta_{i}\right)\pi_{\alpha }d\theta_{i}+\int_{\alpha }^{1}I\left(\theta_{i}\right)\pi_{\alpha }d\theta_{i}\\ & = & \frac{2}{(n-1)\alpha }\sum_{j=1}^{k}\left(p_{ij}-p_{nj}\right)\left[\int_{0}^{\alpha }\frac{\theta_{i}}{n-1}\frac{p_{ij}-p_{nj}}{p_{\theta }^{j}}d\theta_{i}\right]\\ & & +\frac{2}{\left(n-1)(1-\alpha \right)}\sum_{j=1}^{k}\left(p_{ij}-p_{nj}\right)\left[\int_{\alpha }^{1}\frac{1-\theta_{i}}{n-1}\frac{\left(p_{ij}-p_{nj}\right)}{p_{\theta }^{j}}d\theta_{i}\right]\\ & = & \frac{2}{(n-1)\alpha }\sum_{j=1}^{k}\left(p_{ij}-p_{nj}\right)\left[\int_{0}^{\alpha }\left(1-\frac{p_{\left(0_{i},\theta \right)}^{j}}{p_{\theta }^{j}}\right)d\theta_{i}\right]\\ & & -\frac{2}{\left(n-1)(1-\alpha \right)}\sum_{j=1}^{k}\left(p_{ij}-p_{nj}\right)\left[\int_{0}^{\alpha }\left(1-\frac{p_{\left(1_{i},\theta \right)}^{j}}{p_{\theta }^{j}}\right)d\theta_{i}\right]\\ & = & \frac{2}{\alpha }\sum_{j=1}^{k}p_{\left(0_{i},\theta \right)}^{j}\log\left\{\frac{p_{\alpha }^{j}}{p_{\left(0_{i},\theta \right)}^{j}}\right\}+\frac{2}{1-\alpha }\sum_{j=1}^{k}p_{\left(1_{i},\theta \right)}^{j}\log\left\{\frac{p_{\left(1_{i},\theta \right)}^{j}}{p_{\alpha }^{j}}\right\}\\ & = & \frac{2}{\alpha (1-\alpha )}\left[\alpha KL\left(P_{\left(1_{i},\theta \right)},P_{\alpha }\right)+\left(1-\alpha \right)KL\left(P_{\left(0_{i},\theta \right)},P_{\alpha }\right)\right]\\ & = & \frac{2}{\alpha (1-\alpha )}JS\left(P_{\left(0_{i},\theta \right)},P_{\left(1_{i},\theta \right)};\alpha \right), \end{array}$$
as required.  □

## 5. Concluding Remarks

In this paper, we have introduced the discrete version of Jensen–Fisher information measure, and have shown that this information measure can be expressed as a mixture of discrete Fisher information distance measures. Further, we have considered a finite mixture probability mass function and have derived Fisher information and Bayes–Fisher information for the mixing parameter vector. We have shown that the Fisher information for the mixing parameter is connected to chi-square divergence. We have also studied the Bayes–Fisher information for the mixing parameter of a finite mixture model under some prior distributions. These results have provided connections between the Bayes–Fisher information and some known informational measures such as Shannon entropy, Kullback–Leibler, Jeffreys and Jensen–Shannon divergence measures.

## Figures and Tables

**Figure 1 entropy-23-00363-f001:**
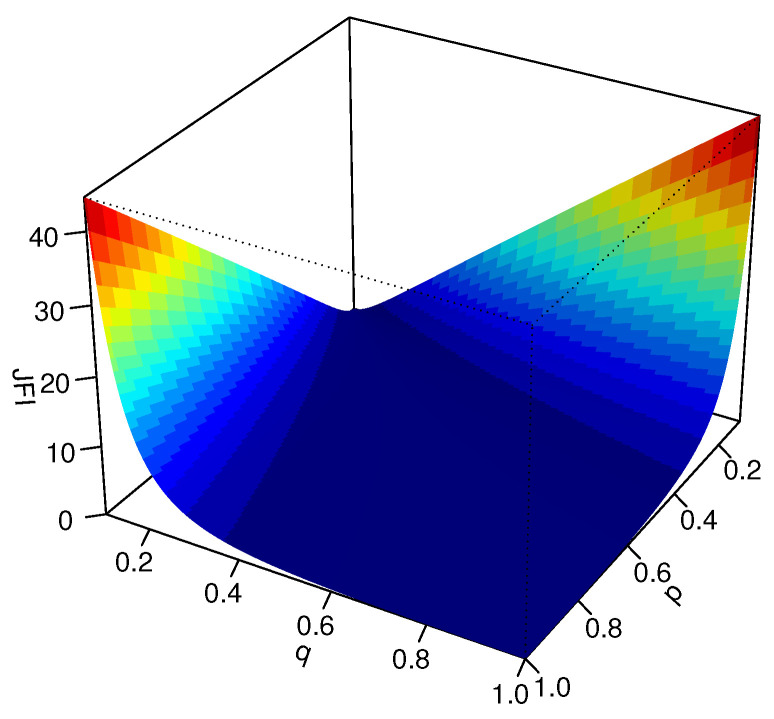
3D-plot of the DJFI divergence between the PMFs P=(p,1−p) and Q=(q,1−q).

## Data Availability

Data sharing not applicable.

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
