# Peer review of "Discrete Versions of Jensen–Fisher, Fisher and Bayes–Fisher Information Measures of Finite Mixture Distributions"

_entropy, 2021, doi:10.3390/e23030363_

Round 1

Reviewer 1 Report

The idea undelying the paper might be of interest, if strongly motivated both by examples and by global statistical issues. Relation between the discrete Fisher information number and its statistical counterpart in standard statistical models is a must.

Author Response

 First of all, we thank the reviewers for their valuable comments which led to an improvement
in the manuscript entitled \Discrete Versions of Jensen-Fisher, Fisher and Bayes-Fisher Information Measures of Finite Mixture Distributions". Based on the to reviewers’ comments, we
have prepared a careful revision of this article. Here is our point-by-point response to reviewers’
comments
.

Reviewer 2 Report

In this paper, the discrete version of Jensen-Fisher information measure is introduced and represented as a mixture of discrete Fisher information distance measures. The Bayes-Fisher information for the mixing parameter of a finite mixture model under some prior distributions is derived and studied. Connections between the Bayes Fisher information and several entropy measures and divergence measures are revealed. 

By extending information theory concepts and deriving their relevant properties, this paper provides a clearer picture of previous results in this field, derives interesting connections and highlights their importance. The results obtained in this framework open research directions for new extensions of other informations measures and therefore I consider this paper can be accepted for publication.

Author Response

(The authors gave the same response as above.)

Reviewer 3 Report

Review of "Discrete Versions of Jensen-Fisher, Fisher and Bayes-Fisher Information Measures of Finite Mixture Distributions" by Kharazmi & Balakrishnan

The authors presented a novel paper with the main properties of discrete version of Fisher information and Jensen-Fisher information measures. They presented the Fisher information for mixing parameter vector of a finite mixture probability mass function, under several priors. I think that the paper has merit to be published in Entropy journal, however some few issues must be addressed by the authors:

1. L2: "information, to develop".
2. L2: delete "for this new divergence measure".
3. L29-30: An extra paragraph with recent developments of entropy for finite mixtures is needed: Upper and lower bounds for gaussian mixture entropy were obtained by Kolchinsky & Tracey (2017), using the Bhattacharyaa distance and Kullback-Leibler divergence. Contreras-Reyes and Cortés (2016) and Abid et al. (2021) provides upper and lower bounds for Shannon and Rényi entropies of non-gaussian finite mixtures, the skew-normal and skew-t cases, respectively.
4. L36: delete "the convention that".
5. L52-53: delete "We first present the definitions of informational measures that are considered in this paper", because it is unnecessary if you defined later the Shannon and KL divergence and, in the previous paragraph (L41), you already mentioned the "first purposes".
6. L53: "Let X be a discrete".
7. Theorem 4: Xi-square function is not defined. It is the usual Xi-square density? In proofs [Eq. (7)], I can see that it could be I(P) function of Eq. (1)?
8. Some plots could be added for the most important results: Definitions 1, 2 and 3. To do this, see pp. 16 of the book of Cover & Thomas (2006), i.e., for two sets p_1,...,p_n and q_1,...,q_n, give 3d-plots for these measures. 

References:

Kolchinsky, A., Tracey, B. D. (2017). Estimating mixture entropy with pairwise distances. Entropy 19(7), 361.

Contreras-Reyes, J.E., Cortés, D.D. (2016). Bounds on Rényi and Shannon Entropies for Finite Mixtures of Multivariate Skew-normal Distributions: Application to Swordfish (Xiphias gladius Linnaeus). Entropy 18(11), 382.

Abid, S.H., Quaez, U.J., Contreras-Reyes, J.E. (2021). An information-theoretic approach for multivariate skew-t distributions and applications. Mathematics 9(2), 146.

Cover, T., Thomas, J. (2006). Elements of Information Theory. Wiley, 2nd ed.

Author Response

 First of all, we thank the reviewers for their valuable comments which led to an improvement
in the manuscript entitled \Discrete Versions of Jensen-Fisher, Fisher and Bayes-Fisher Information Measures of Finite Mixture Distributions". Based on the to reviewers’ comments, we
have prepared a careful revision of this article. Here is our point-by-point response to reviewers’
comments

Round 2

Reviewer 1 Report

Thank you for revision

Reviewer 3 Report

2nd review of "Discrete Versions of Jensen-Fisher, Fisher and Bayes-Fisher
Information Measures of Finite Mixture Distributions" by Kharazmi and Balakrishnan (2021)

In this 2nd review I can see that authors have addressed all of my previous comments/suggestions of my 1rst review. However, I have some questions about new figure 3:

  1. If this is Fig. 3, where is the previous figures 1 and 2?
  2. Fig. 3 looks rough, I propose to authors to consider the following R code to give a smooth aspect of this figure:

library(GA)

p <- q <- seq(0.1,1,0.005)
n = length(p)
JFI <- matrix(NA, n, n)
J <- function(i,j) 0.5*(i+j)*((1-i)/i - (1-j)/j)^2

for(i in 1:n) for(j in 1:n) JFI[i,j] = J(p[i], q[j])

persp3D(p, q, JFI, phi = 30, theta = 120, box = TRUE, 
        shade = .1, expand=0.75)

3. Finally, put some comments about this figure: JFI incremments when probabilities increase, and JFI is maximized when p is close to cero and q is close to 1, and visiversa. However, if p and q are simultaneously close to 0, JFI is close to 0.